# Formulated Phospholipids as Non-Canonical TLR4 Agonists

**DOI:** 10.3390/pharmaceutics14122557

**Published:** 2022-11-22

**Authors:** Hong Liang, William R. Lykins, Emilie Seydoux, Jeffrey A. Guderian, Tony Phan, Christopher B. Fox, Mark T. Orr

**Affiliations:** 1Access to Advanced Health Institute (AAHI), 1616 Eastlake Ave E, Suite 400, Seattle, WA 98102, USA; 2Department of Global Health, University of Washington, 3980 15th Ave NE, Seattle, WA 98195, USA

**Keywords:** oil-in-water emulsions, phospholipids, adjuvants, Toll-like receptor agonists

## Abstract

Immunogenic agents known as adjuvants play a critical role in many vaccine formulations. Adjuvants often signal through Toll-like receptor (TLR) pathways, including formulations in licensed vaccines that target TLR4. While TLR4 is predominantly known for responding to lipopolysaccharide (LPS), a component of Gram-negative bacterial membranes, it has been shown to be a receptor for a number of molecular structures, including phospholipids. Therefore, phospholipid-based pharmaceutical formulations might have off-target effects by signaling through TLR4, confounding interpretation of pharmaceutical bioactivity. In this study we examined the individual components of a clinical stage oil-in-water vaccine adjuvant emulsion (referred to as a stable emulsion or SE) and their ability to signal through murine and human TLR4s. We found that the phospholipid 1,2-dimyristoyl-sn-glycero-3-phosphocholine (DMPC) activated TLR4 and elicited many of the same immune phenotypes as canonical TLR4 agonists. This pathway was dependent on the saturation, size, and headgroup of the phospholipid. Interestingly, DMPC effects on human cells were evident but overall appeared less impactful than emulsion oil composition. Considering the prevalence of DMPC and other phospholipids used across the pharmaceutical space, these findings may contextualize off-target innate immune responses that could impact preclinical and clinical development.

## 1. Introduction

Adjuvanted protein vaccines represent a large and growing fraction of the vaccine marketplace, including but not limited to approved products from Dynavax (the Heplisav-B vaccine against hepatitis B) and GSK (the Shingrix vaccine against shingles) and products in clinical development against severe acute respiratory syndrome coronavirus 2 (SARS-CoV-2) from Novavax and Sanofi-GSK among others [1,2,3]. The immunogenicity of adjuvanted protein vaccines depends on molecular agents known as adjuvants, which stimulate the innate immune system to program a more effective adaptive immune response to the delivered protein antigen [4]. Three broad classes of adjuvants are currently used with licensed human vaccines: aluminum-containing adjuvants (e.g., Alhydrogel and AS04), oil-in-water emulsions (e.g., MF59 and AS03), and Toll-like receptor (TLR) agonists (e.g., AS01b and CpG-1018), often formulated with aluminum or lipid-based particles [5]. Aluminum and emulsion adjuvants primarily augment antibody responses to vaccine antigens by elongating germinal center exposure, whereas the combination adjuvants with TLR agonists also promote CD4^+^ T cell responses [5,6]. In the context of SARS-CoV-2, TLR-agonist adjuvanted peptide vaccines that can elicit both CD4^+^ and CD8^+^ T cell responses were found to be effective against a wide range of emerging variants [7].

TLRs are pattern recognition receptors (PRRs) capable of recognizing specific pathogen-associated molecular patterns (PAMPs) conserved among micro-organisms and stimulate inflammatory signaling cascades. Since the discovery of TLR pathways in the 1990s and the identification of lipopolysaccharide (LPS) as the canonical agonist of the TLR4 pathway, numerous studies have identified alternative agonists and antagonists [8,9,10,11,12,13]. These alternate signaling molecules can be small molecules, LPS mimics, or a range of other structures. Additionally, compounds have been identified that act on up- or downstream components of the TLR4 signaling pathway, which are often derived from metabolic byproducts but can also be exogenously derived [13]. This has led to a library of hundreds to potentially thousands of molecular agents with known activity in the TLR4 pathway. Substantial progress has been made in the last several years in defining the mechanisms of action for many of these compounds. This knowledge has enabled rational development of next-generation adjuvants and more informative clinical evaluation of new adjuvanted vaccines [14]. Among their potential mechanisms of action, adjuvants can act on antigen-presenting cells (APCs) through inflammasome and/or TLR-mediated pathways. Inflammasome activation via TLR4 requires two signals: First, signal 1 engages TLR4 through the adaptor proteins myeloid differentiation factor 2 (MD-2), LPS-binding protein, and CD14 to activate nuclear factor-κB (NF-κB) to produce nucleotide-binding domain-like receptor protein 3 (NLRP3) and pro-interleukin-1β (pro-IL-1β). Next, signal 2 stimulates NLRP3, pro-caspase-1, and the adaptor protein apoptosis-associated speck-like protein containing a caspase activation and recruitment domain (ASC) assembly, which subsequently cleaves the pro-caspase into its active form caspase-1, which then cleaves pro-IL-1β and pro-IL-18 (constitutively expressed) into active, secretory forms [15,16].

Glucopyranosyl lipid adjuvant (GLA) is a synthetic TLR4 agonist that, when formulated with a phospholipid-stabilized squalene-in-water stable emulsion (SE), forms a safe and effective vaccine adjuvant, which has advanced to Phase 2 clinical testing [17,18]. The combination adjuvant GLA-SE promotes strong TH1 cellular and balanced IgG1/IgG2 antibody responses to a variety of vaccine antigens in animal models and human clinical studies and provides protective immunity against infections such as tuberculosis and leishmaniasis [15,19]. The adjuvanticity of GLA is critically dependent on its formulation in SE [20]. We have found that GLA-SE mediates TH1 induction via myeloid differentiation factor 88 (MyD88) and Toll/IL-1 receptor domain-containing adaptor protein inducing interferon-β (TRIF) signaling and produces IL-18 in draining lymph nodes, suggesting the inflammasome is involved [21,22]. We have further demonstrated that SE induces APC recruitment in the draining lymph nodes, which drives the development of adaptive immunity. The adjuvanticity of SE was substantially impaired in ASC^−/−^ or NLPR3^−/−^ mice, suggesting that SE functions in an inflammasome-dependent manner [15]. These findings further support a two-step mechanism of action for the combination GLA-SE adjuvant to activate the inflammasome in which (1) GLA, signaling via TLR4, increases expression of the inflammasome components NLRP3 and ASC and the inflammasome substrates pro-IL-1β and pro-IL-18, and (2) SE activates the NLRP3-dependent inflammasome to activate caspase-1 to process the proenzymes into their secreted active forms.

Despite progress, the specific mechanisms by which oil-in-water emulsions engage with the innate immune system are still not completely understood. Recently, fatty acids and oxidized phospholipids that result from cellular damage have been shown to engage with TLR4-dependent pathways in animal models in a pro- or anti-inflammatory manner [12,23,24,25]. These damage-associated molecular patterns (DAMPs) often result from the partial oxidation of unsaturated fatty acids such as 1-palmitoyl-2-arachidonoyl-phosphatidylcholine (PAPC), leading to a heterogeneous pool of products (i.e., oxidized PAPC [oxPAPC]) [26]. oxPAPC in particular has been shown to block LPS activity in murine models of septic shock by competitively inhibiting LPS binding to CD14 and preventing downstream signaling [23,27]. Further, pre-treatment with oxPAPC can even reduce mouse survival against peritoneal *Escherichia coli* challenge [28]. However, oxPAPC can also induce a prolonged hyper-inflammatory state when dosed after LPS [25]. Additionally, several recent reports have suggested that phosphatidylcholine (PC) molecules may interact with TLR4 [23,24,25].

Other phospholipids, such as those present in adjuvant emulsions like SE, may also engage with the TLR4 pathway. SE consists of a squalene-in-water emulsion stabilized by non-ionic surfactants and emulsifiers, including poloxamer 188 and the phospholipid 1,2-dimyristoyl-sn-glycero-3-phosphocholine (DMPC), and has been evaluated in a number of clinical-stage vaccine models [29,30,31]. In this report, we show that DMPC present in SE can stimulate the TLR4 pathway. We demonstrate that DMPC is necessary and sufficient to recapitulate the TLR4 pathway-stimulating effect of SE, and can stimulate the production of antigen-specific antibodies in a TLR4-dependent manner. Additionally, we observed that phospholipid-stimulated IL-1β production in murine bone marrow-derived dendritic cells (BMDCs) was sensitive to the chemical composition of both the phospholipid tail and headgroups. The results of our study may be relevant to the development of phospholipid-containing delivery systems and their characterization in preclinical and clinical models.

## 2. Materials and Methods

### 2.1. Reagents

SE (5X concentrate consisting of 10% *v*/*v* squalene, 1.9% *w*/*v* DMPC, 0.09% *w*/*v* poloxamer 188, 1.8% *v*/*v* glycerol, and 25 mM ammonium phosphate buffer) was made in-house by high-pressure homogenization as previously described [20]. Grapeseed SE (2X concentrate consisting of 4% *v*/*v* grapeseed oil, 0.76% *w*/*v* DMPC, 0.036% *w*/*v* poloxamer 188, 1.8% *v*/*v* glycerol, and 25 mM ammonium phosphate buffer) was prepared by the same high-pressure homogenization method as above. MF59-like emulsion (2X concentrate consisting of 4% *v*/*v* squalene, 0.4% *w*/*v* sorbitan trioleate, 0.4% *w*/*v* polysorbate 80, and 10 mM citrate buffer) was manufactured by high-pressure homogenization as described previously for AAHI’s MF59-like EM022 composition [32]. Liposomes containing one of the following PCs were formulated at 1.9% (*w*/*v*) in deionized water: 1-palmitoyl-2-oleoyl-sn-glycero-3-phosphocholine (POPC); 1,2-dioleoyl-sn-glycero-3-phosphocholine (DOPC); 1,2-distearoyl-sn-glycero-3-phosphocholine (DSPC); 1,2-dipalmitoyl-sn-glycero-3-phosphocholine (DPPC); DMPC; 1,2-dilauroyl-sn-glycero-3-phosphocholine (DLPC); and 1,2-dimyristoyl-sn-glycero-3-phospho-(1’-rac-glycerol) (DMPG) were obtained from Lipoid (Newark, NJ, USA) or Avanti Polar Lipids (Alabaster, AL, USA). 1,2-dimyristoyl-sn-glycero-3-phosphoethanolamine (DMPE) and 1,2-dimyristoyl-sn-glycero-3-phospho-L-serine (DMPS) were obtained from Sigma Aldrich (St. Louis, MO, USA). The liposome mixture was then bath-sonicated at 65 °C for 20–60 min followed by 0.2-µm filtration. Particle size was determined by diluting an aliquot of each formulation 1:100-fold in deionized water and measuring the scattering intensity-based size average (*Z*-avg) by dynamic light scattering with a Zetasizer APS (Malvern Panalytical, Malvern, UK). Particle size and polydispersity index for materials used in this study can be found in Appendix A. GLA-AF (5X concentrate consisting of 0.25 mg/mL GLA and 0.21 mg/mL DPPC) was manufactured in-house by sonication as previously described [20]. Ultrapure LPS (*E. coli* 0111:B4), FSL-1, and polymyxin B were obtained from InvivoGen (San Diego, CA, USA). Adenosine 5′-triphosphate (ATP) was purchased from Thermo Fisher Scientific (Waltham, MA, USA).

### 2.2. Animal Ethics

Female C57BL/6 wild-type (WT), TLR2^−/−^, TLR4^−/−^, MyD88^−/−^, and NLRP3^−/−^ mice aged 6–10 weeks were purchased from The Jackson Laboratory (Harbor, ME, USA). All strains were maintained in specific-pathogen-free (SPF) conditions. All animal experiments and protocols used in this study were approved by the Infectious Disease Research Institute, now AAHI, Institutional Animal Care and Use Committee (IACUC) and the Office of Laboratory Animal Welfare Assurance (Assurance ID A4337-01) effective 25 February 2015, to 28 February 2019.

### 2.3. Bone Marrow-Derived Dendritic Cell Prime and Stimulation In Vitro

BMDCs derived from WT or TLR2^−/−^, TLR4^−/−^, MyD88^−/−^, or NLRP3^−/−^ mice were cultured following the protocol developed by Lutz et al. [33]. Briefly, BMDCs were derived from bone marrow and allowed to differentiate in IMDM (Iscove’s Modified Dulbecco’s Medium) containing 10% fetal bovine serum (FBS), 1% L-glutamine, 1% penicillin/streptomycin, and 20 ng/mL granulocyte-macrophage colony-stimulating factor (GM-CSF). Non-adherent cells were collected between Day 8 and Day 10 for stimulation. Collected cells were centrifuged and resuspended in OptiPRO SFM (Thermo Fisher Scientific). 125,000 cells in 125 μL were plated per well in a 48-well plate. Cells were primed by treatment with LPS (10 μg/mL) or media for 2 h then re-centrifuged prior to treatment as indicated below immediately after plating and incubated at 37 °C with 5% CO_2_ for 5 h: SE or MF59-like emulsion with squalene or grapeseed oil at 0.5% oil (*v*/*v*), phospholipid and liposomes at 0.095% (*w*/*v*), LPS (TLR4 agonist) at 10 μg/mL, and FSL-1 (TLR2/6 agonist) at 100 ng/mL. Following treatment, cells were centrifuged and supernatant was collected and stored at −20 °C until analysis. To recover cell lysates, 125 μL of 1X radioimmunoprecipitation assay (RIPA) buffer with protease inhibitors (Roche, Basel, Switzerland) was added per well and allowed to incubate as per the manufacturer’s instructions prior to the collection of lysates. Potential LPS contamination was assessed by pre-treating LPS, SE, and DMPC liposomes with 50 µg/mL polymyxin B for 5 h before addition to BMDC cultures as described above.

### 2.4. Cytokine Enzyme-Linked Immunosorbent Assays (ELISAs)

To enable qualitative comparison between cytoplasmic (pro-IL-1β) and secreted IL-1β, we chose to use ELISA as opposed to a more direct cellular method, such as intracellular cytokine staining. Supernatants and lysates from primary BMDC assays were assessed for production of IL-1β (supernatant), pro-IL-1β (lysate), IL-18 (lysate), tumor necrosis factor alpha (TNFα, lysate), and IL-12p40 (lysate) via ELISA (Thermo Fisher Scientific) according to the manufacturer’s instructions. Cytokine quantification was performed via a standard curve using GraphPad Prism v9 software (San Diego, CA, USA).

### 2.5. Reporter Cell Assays

HEK-Blue mTLR4, hTLR4, and mTLR7 reporter cells were obtained from InvivoGen (San Diego, CA, USA). For these cells, a secreted embryonic alkaline phosphatase (SEAP) reporter gene is placed under the control of an interferon (IFN)-β minimal promoter fused to five NF-κB and AP-1 binding sites. Stimulation with a TLR4 agonist activates NF-κB and AP-1, which induce the production of SEAP. The production of SEAP induced by NF-κB and AP-1 activation, which is triggered by TLR4 or TLR7 stimulation, is measured at an optical density (OD) of 650 after 6 h following the protocol provided by the manufacturer (InvivoGen). Cells were cultured at 25,000 cells/well and treated with either DMPC (150 μg/mL), GLA-AF (50 μg/mL), 3M-052-AF (1 μg/mL), or media, and incubated at 37 °C for 40 h (hTLR4 and mTLR4) or 44 h (mTLR7). Each stimulation condition was performed in triplicate.

### 2.6. Human Whole Blood Immunogenicity Assay

SE and MF59-like emulsions comprising squalene or grapeseed oil were evaluated for innate immunostimulatory activity on whole blood from 6 healthy human subjects (3 male and 3 female). The emulsions were incubated directly with heparinized whole blood at 0.4% (*v*/*v*) oil for 18–24 h at 37 °C, and production of monocyte chemoattractant protein-1 (MCP-1), IL-8, and macrophage inflammatory protein-1β (MIP-1β) cytokines in supernatants was quantified as described previously [34].

### 2.7. Mice and Immunizations

Female WT and TLR4^−/−^ mice were immunized via an intramuscular injection in the quadriceps muscles of hind limbs (50 µL per leg) with formulations containing 2.5 µg of recombinant ID97 tuberculosis antigen formulated with a squalene-based MF59-like emulsion or SE both at 2% oil by volume [35]. Blood was collected via terminal cardiac bleeding on Day 21, and serum was isolated prior to storage and analysis.

### 2.8. Serum Endpoint Titer ELISA

Serum titers against ID97 antigen were evaluated by antibody-capture ELISA. Briefly, Corning high-binding 384-well plates (VWR International, Radnor, PA, USA) were coated overnight at 4 °C with 2 µg/mL ID97 in coating buffer (eBioscience, San Diego, CA, USA), then washed in phosphate-buffered saline (PBS)-Tween 20. Serially diluted serum samples were incubated for 1 h followed by either anti-mouse IgG (H + L)-HRP, IgG1-HRP, or IgG2c-HRP (SouthernBiotech, Birmingham, AL, USA); and 3,3′,5,5′-tetramethylbenzidine (TMB) substrate was applied as per the manufacturer’s instructions. Plates were analyzed at 450 nm using an ELx808 Absorbance Reader (Bio-Tek Instruments, Winooski, VT, USA), and endpoints were set as the minimum dilution at which values were lesser than or equal to an OD of 0.5.

### 2.9. Statistical Analysis

Data were analyzed using GraphPad Prism by one-way ANOVA (with corrections for multiple comparisons applied as indicated in figure captions). Values were considered significantly different with *p* < 0.05 (*), *p* < 0.01 (**), *p* < 0.001 (***), and *p* < 0.0001 (****).

## 3. Results

### 3.1. SE Does Not Induce Mature IL-1β Secretion through the Inflammasome

To determine whether SE activates the inflammasome, we treated media or LPS-primed murine BMDCs with SE or the known inflammasome activator ATP [36]. LPS prime and ATP treatment resulted in secretion of mature IL-1β into the supernatant, whereas SE treatment did not cause extracellular IL-1β accumulation (Figure 1A). To our surprise, treatment of naïve, i.e., not LPS-primed, BMDCs with SE caused an accumulation of intracellular pro-IL-1β, similar to LPS priming. The accumulation of pro-IL-1β in the lysate was significantly greater in SE-treated cells compared to untreated or ATP-treated cells. This suggests that SE, or one of its components, can activate signal 1, but not signal 2, of the inflammasome pathway, similar to LPS.

### 3.2. TLR4 and MyD88 Are Crucial for the Pro-IL-1β Induction Activity

Most TLRs, except TLR3, depend on the MyD88 adaptor protein to effectively link PAMP recognition to changes in gene expression and to mediate signal 1 of inflammasome activation. Therefore, to understand if SE was specifically interacting with known TLR4 signaling pathways, we employed transgenic MyD88^−/−^ animals and cell lines as negative controls [37]. Consistent with SE being a canonical TLR4 agonist, pro-IL-1β expression was ablated in MyD88^−/−^ BMDCs treated with GLA or SE, whereas the inflammasome molecule NLRP3 was not required for pro-IL-1β expression (Figure 1B). To determine which MyD88-associated receptors recognized SE, we examined responses in TLR4- and TLR2-deficient BMDCs. TLR4^−/−^, but not TLR2^−/−^, BMDCs failed to increase pro-IL-1β expression in response to SE treatment, indicating that SE acts on the TLR4-MyD88 signaling axis (Figure 1C). Importantly, TLR4^−/−^ BMDCs produced normal amounts of pro-IL-1β in response to the TLR2/6 ligand FSL-1, confirming that these cells are not globally defective in pro-IL-1β expression (Figure 1D). This suggests that some component of SE is stimulating pro-IL-1β production specifically via a TLR4 and MyD88-mediated process.

### 3.3. DMPC, a Component of SE, Is Essential for TLR4 Activity

To determine whether squalene or DMPC is signaling through the TLR4-MyD88 pathway to enhance pro-IL-1β expression, we developed stable oil-in-water emulsions lacking either DMPC or squalene and applied them to murine BDMCs. We have previously found that replacing squalene with other oils, including grapeseed oil, impairs the adjuvant activity of oil-in-water emulsions, allowing us to use grapeseed oil as a non-immunogenic control [38]. Squalene is also a primary component of both the MF59 and AS03 emulsions used in licensed vaccine products. Therefore, an MF59-like emulsion was used as a non-DMPC-containing control for the immunogenicity of squalene. GLA in an aqueous formulation (GLA-AF) was used as a positive control for TLR4 stimulation. The DMPC-containing grapeseed oil SE retained pro-IL-1β activity, whereas the DMPC-free MF59-like emulsion was not active (Figure 2A). Treatment of BMDCs with DMPC formulated as a liposome without squalene was also sufficient to elicit pro-IL-1β production and other pro-inflammatory cytokines including IL-18, TNFα, and IL-6 (Figure 2B). DMPC liposomes and DMPC-containing emulsions were able to elicit inflammatory signals similar to the canonical agonist GLA, suggesting an equivalent immune phenotype.

### 3.4. DMPC Is an Agonist of the Murine and Human TLR4 Pathways

To further validate the TLR4 activity of DMPC, we employed a TLR4^−/−^ mouse model and a transgenic reporter model for both mouse and human TLR4 activity in a human cell line. As with SE, pro-IL-1β production by DMPC liposome stimulation was ablated in TLR4^−/−^ BMDCs (Figure 3A). These results suggest that DMPC plays an essential role in SE TLR4 activity. However, LPS (endotoxin) is a TLR4 ligand, and its contamination in test reagents could generate false conclusions regarding TLR4 ligands. Polymyxin B is able to selectively bind LPS and neutralize its biological activity [39]. Therefore, to confirm that SE and DMPC are bona fide TLR4 agonists with pro-IL-1β activities and not the result of endotoxin contamination, we treated LPS, SE, and DMPC liposomes with 50 µg/mL polymyxin B before adding them to BMDCs. Addition of polymyxin B was sufficient to abrogate the pro-IL-1β response to LPS but had no impact on the response to DMPC or SE, indicating that their activities are not due to endotoxin contamination (Figure 3B).

Murine and human TLR4s have different specificities for some ligands [40,41]. To determine whether DMPC engages the human TLR4, as well as murine TLR4, we employed HEK-Blue reporter cells transfected with either murine TLR4 (and species-matched adaptor proteins MD-2 and CD14) or human TLR4. Both human and murine TLR4 reporters were strongly responsive to GLA-AF stimulation as a canonical positive control and moderately responsive to DMPC stimulation, whereas a TLR7 reporter system was non-responsive to DMPC but was activated by the known TLR7/8 agonist 3M-052 as a positive control (Figure 3C) [20]. Other studies using HEK-Blue cells overexpressing human or murine TLR4 have shown similar responses to DMPC stimulation [40,42]. These results demonstrate that DMPC is the TLR4 active component of SE, that activity is not due to endotoxin contamination, and that the activity of DMPC is maintained across both murine and human TLR4s.

### 3.5. Emulsion Stimulation of Innate Immune Activity in Human Whole Blood Is Dominated by Oil-Phase Components

Adjuvant immunogenicity varies between species, and the stimulation of chemokine secretion to attract innate immune cell populations is a critical step toward adaptive immunity [8,43,44,45]. To further understand the activity of DMPC on markers of human innate immunity, we applied SEs and MF59-like emulsions formulated from either squalene or grapeseed oil to human whole blood from 6 donors (3 male and 3 female) for 18–24 h and assayed a panel of secreted cytokines via ELISA. Monocyte chemoattractant protein-1 (MCP-1 or CCL2), IL-8, and macrophage inflammatory protein-1β (MIP-1β or CCL3) were chosen because of their role in early innate immune cell recruitment and because of their previous use in other preclinical vaccine formulation studies [34,46,47]. The squalene-containing formulations (SE and MF59-like emulsion) were more effective at eliciting MCP-1 (Figure 4A) and MIP-1β (Figure 4B) secretion than the grapeseed-containing formulations, regardless of DMPC content. However, the DMPC-containing grapeseed SE stimulated the secretion of more IL-8 than the grapeseed oil MF59-like emulsion (Figure 4C), confirming a TLR4/MyD88/NF-κB-dependent IL-8 secretion previously reported and further validating DMPC as a TLR4 ligand [48,49]. For some markers of human immunogenicity such as production of IL-8, DMPC clearly has some effect when compared to the DMPC-free MF59-like emulsions. However, oil selection (e.g., squalene vs. grapeseed oil) has a more pronounced phenotype.

### 3.6. Phospholipid Acyl Chain Length, Saturation State, and Headgroup Structure Are Critical for TLR4 Activity

DMPC contains a PC headgroup and two saturated acyl chains, each containing 14 carbons (14:0). To determine the structural motifs of DMPC that are important for TLR4 engagement, we examined the pro-IL-1β response of BMDCs stimulated with liposomes comprising structural variants of phospholipids with different headgroups or acyl chains (Figure 5A). DLPC, which has the same headgroup as DMPC but shorter acyl chains (C12, 12:0), showed similar pro-IL-1β production activity to DMPC (Figure 5B). Conversely, longer acyl chains with the same phosphocholine headgroup as DMPC, either the fully saturated C16 (16:0) DPPC or C18 (18:0) DSPC, the monounsaturated C18 (18:1) DOPC, or the asymmetrical C16/C18 (16:0/18:1) POPC, all elicited very little pro-IL-1β compared to DMPC (Figure 5B). Among the three saturated C14 phospholipids, DMPC and DMPG induced the production of pro-IL-1β whereas DMPS did not, suggesting that the headgroup structure also impacts TLR4 engagement (Figure 5B). This demonstrates that both acyl chain length and headgroup structure determine the immunogenicity of phospholipids in murine BMDCs.

### 3.7. TLR4 Plays a Role in Murine SE-Stimulated Antibody Responses

Finally, to determine if TLR4 engagement by the DMPC-containing SE adjuvant contributes to its ability to augment the adaptive immune response, we immunized WT and TLR4^−/−^ C57BL/6J mice with ID97, a protein tuberculosis antigen, adjuvanted with squalene SE or squalene MF59-like emulsion. Three weeks after immunization, WT mice who received the SE-adjuvanted vaccine showed significantly enhanced serum antigen-specific IgG and IgG1 antibody production compared to TLR4^−/−^ mice who received the same vaccine (Figure 6A,B). In mice that received the SE adjuvant, IgG2c production was not affected by TLR4 knockout (Figure 6C). These differences between WT and TLR4^−/−^ mice were not observed using the MF59-like emulsion, which does not contain DMPC, suggesting that SE acts through TLR4 to promote these antibody responses and that DMPC is the TLR4 active compound (Figure 6A,B). This suggests that the adjuvanting effect of SE is related to the inclusion of DMPC.

## 4. Discussion

In this study, we present evidence that DMPC functions as a TLR4 agonist in the context of the SE adjuvant. We show that DMPC can trigger signal 1 of the TLR4-dependent inflammasome pathway and can induce an immune phenotype similar to canonical TLR4 agonists such as GLA. We additionally show that DMPC can engage with both human and murine TLR4s, although the immunogenicity of DMPC is overshadowed by the impacts of oil selection. Finally, we demonstrate that phospholipid structure impacts its ability to interact with the TLR4 pathway and that DMPC promotes antibody production in adjuvanted protein vaccines in a TLR4-dependent process. Based on our results clarifying the pathway interactions in Figure 1 and our results showing a dependence on chemical structure in Figure 5, we believe that there is a specific molecular interaction between DMPC and TLR4 mediated by MyD88.

Based on previous studies of other lipids associated with TLR4 engagement, DMPC may be stabilized in the MD-2 binding pocket and facilitate the association between MD-2 and TLR4, leading to downstream signaling through MyD88 and/or TRIF [50,51]. This conclusion is supported by the induction of pro-IL-1β production in murine BMDCs after treatment with DMPC-containing SE formulations (regardless of oil composition) (Figure 2A) and by the lack of IL-1β expression after SE or DMPC liposome treatment in TLR4^−/−^ murine BMDCs (Figure 1C,D and Figure 3A). The DMPC-containing grapeseed oil emulsion also stimulates more secretion of IL-8 in human whole blood than the grapeseed oil-based MF59-like emulsion (Figure 4C), demonstrating some impact of DMPC on human immunogenicity. Overall, the results presented in Figure 1, Figure 2 and Figure 3 suggest that both DMPC-containing emulsions (SE) and DMPC liposomes can signal through TLR4. These results are distinct from previous findings by other groups looking at the TLR4 activity of long-chain saturated fatty acids, such as palmitic acid, due to the inherit spatial structure of an intact phospholipid [52]. Our findings suggest that DMPC engages the TLR4-MyD88 axis but functions in an inflammasome-independent manner as shown by its resilience against NLRP3 knockout (Figure 1B). This supports the conclusion that DMPC is able to trigger TLR4 signal 1, leading to pro-IL-1β production, but is unable to trigger mature IL-1β secretion. Although inflammasome signaling is a canonical pathway by which some adjuvants trigger antibody production, other studies have indicated that TLR4-agonist emulsions lead to strong humoral immune responses through alternative pathways dependent on IL-18 and subcapsular macrophages [22,53]. Because of the limited ability of SE to provide signal 2 of the canonical inflammasome pathway and its independence from NLRP3 (Figure 1B), SE may be functioning via one of these non-canonical processes.

It is important to consider the type of immune response generated by adjuvant systems when evaluating their applied use. This is often accomplished through the use of representative cytokine production panels. We show that DMPC liposomes can elicit similar levels of the pro-inflammatory cytokines TNFα, IL-6, and IL-1β, and the canonical TH1 signal IL-12p40, as SE and GLA (Figure 2B). DMPC can also elicit equivalent cytokine production independent of oil-phase composition (Figure 2A). We have previously shown that oil selection can impact in vivo immunogenicity and that squalene outperforms other common oils in the context of antigen-specific IgG production and long-lived antibody-secreting plasma cell development [38]. The equivalence in pro-IL-1β production we observed between the squalene and grapeseed oil groups (Figure 2A) suggests the lysate pro-IL-1β is primarily due to DMPC signaling through the TLR4 pathway.

Well-known and characterized differences exist between the binding specificity of murine and human TLR4 pathway adaptors, which potentially stifles the translation of results in small animal models to human clinical products [41,54]. While we observed similar IL-1β responses in HEK cells overexpressing murine or human TLR4 after treatment with DMPC (Figure 3C), we only saw effects of DMPC on IL-8 production in human whole blood (Figure 4) and instead found that oil-phase lipid (e.g., squalene or grapeseed oil) dominated the pro-inflammatory cytokine response (MCP-1 and MIP-1β). This suggests that while DMPC may partially engage the human TLR4 pathway, it does not seem to play a substantial role in human immunogenicity. Due in part to the differences in binding pockets between human and murine adaptor proteins (MD-2, TRIF, etc.), agents that are known to be human TLR4 antagonists can be agonists of the murine pathway [40,45]. Therefore, it is unclear at present the manner by which DMPC is interacting with the human TLR4 pathway.

Our results (Figure 5B) further suggest that phospholipid structure may impact the agonist activity of adjuvant systems. Previous studies have shown that saturated lipid chains 10–14 carbons in length are the most effective at activating or suppressing TLR4 signaling and that bioactivity drops off dramatically with longer aliphatic chains and degree of unsaturation [13,33]. We observed that fully saturated 12–14 carbon PC lipids, but not unsaturated or longer chain structures, stimulate pro-IL-1β production in murine BMDCs. In contrast, we have shown that a number of phospholipid-containing emulsions, including both TLR4-active (DMPC) and TLR4-inactive (DOPC, POPC) PCs, have similar adjuvanting properties in mice [55]. This suggests that while DMPC has specific activity in the TLR4 pathway, the downstream biological implications of this are secondary to the adjuvanting effect of intact emulsion formulations [20,56]. Interestingly, DMPS and DMPG liposomes exhibit dramatically different behavior in terms of pro-IL-1β production, despite having identical aliphatic portions and similarly sized headgroups. It is unclear if this is due to DMPS functioning as an antagonist while still engaging with the same binding pocket or if it is entirely precluded from interacting with the TLR4 pathway due to steric hindrance or clashing between charged groups, among other possibilities. It is also unclear if DMPG, which was found to stimulate pro-IL-1β production in murine cells, would have a similar agonistic effect in human models.

Previous studies of the oxidized PC lipid oxPAPC in mice have shown that it interreacts with the TLR4 pathway as an antagonist or agonist depending on the immune context and LPS co-dosing schedule [23,24,25]. When dosed prior to LPS exposure, oxPAPC can competitively inhibit LPS and sepsis, but when dosed after LPS priming, oxPAPC treatment can lead to prolonged hyperinflammation [23,25,27]. This suggests that immune context has a strong influence on the downstream impact of TLR4-phospholipid interactions. This should be considered when developing lipid-based formulations for biologics or other potentially immunogenic compounds (e.g., polyethylene glycol, polylactic acid, etc.), which might lead to undesirable inflammatory responses or adaptive immunity [57,58].

DMPC-containing SE was found to stimulate the production of antigen-specific antibodies against the tuberculosis antigen ID97 in a mouse model via a TLR4-dependent process (Figure 6) [17]. Further, based on the titer ratio of IgG1 (TH2) and IgG2c (TH1), we observed that both the MF59-like and the SE formulation produced a TH2-biased immune response in mice, whereas GLA-SE is known to induce a more TH1-biased response [59]. Unlike the SE treated mice, the antibody response to MF59-like emulsions was unchanged in the TLR4^−/−^ animals. This is in agreement with previous work showing that MF59 acts in a TLR4-independent, although MyD88-dependent, manner [60]. Interestingly, it has been shown previously that only the complete MF59 adjuvant, and not any individual component, was able to augment antibody responses, suggesting that the particulate formulation presentation in particular is the key to its immunogenicity, emphasizing the need for development of composition and formulation in tandem [56]. Conversely, with SE we observed that the removal of TLR4 specifically impacts antibody production, implying that some TLR4-dependent interaction is promoting adaptive immunity. The TLR4-dependent activity of SE and DMPC has potential implications in the development of drug and vaccine formulations. DMPC is a common component of liposomal and emulsion systems used to deliver small molecules and biologics in preclinical and clinical settings. The potential TLR4 activity of selected phospholipids such as DMPC should be taken into account in preclinical study development and interpretation of the biological activity of drug and vaccine formulations.

## 5. Conclusions

The contribution of individual formulation components towards innate and adaptive immunity of SE, a squalene-phospholipid emulsion, was evaluated. Compared to an MF59-like emulsion, which does not contain the phospholipid DMPC, SE was found to act in a TLR4-dependent manner, with DMPC being the key pathway agonist. These conclusions were confirmed in in vitro models of murine and human TLR4 and in vivo murine models of TLR4 activity. We also showed that phospholipids similar to DMPC seem to interact with the TLR4 pathway in a chemical structure-dependent manner. These results contribute to the growing body of literature on the mechanisms of action of adjuvant-emulsion and the interactions between formulation components and the innate immune system, and should be considered in the interpretation of preclinical formulation results. 

## Figures and Tables

**Figure 1 pharmaceutics-14-02557-f001:**
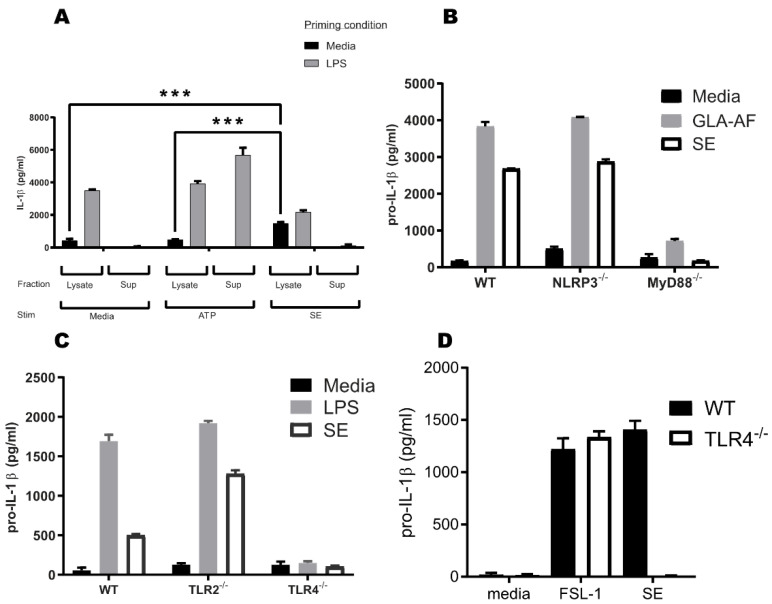
Stable emulsion (SE) elicits pro-IL-1β production in bone marrow-derived dendritic cells (BMDCs) via a TLR4-dependent process. (**A**) SE stimulates signal 1 of the TLR4 inflammasome pathway in the absence of lipopolysaccharide (LPS). BMDCs from wild-type (WT) mice were primed with media or LPS (10 µg/mL) for 2 h and then stimulated with media, 5 mM ATP, or 0.5% SE. Supernatant (sup) and lysate were collected and assayed for IL-1β or pro-IL-1β, respectively, by enzyme-linked immunosorbent assay (ELISA). (**B**) SE stimulates pro-IL-1β production in a MyD88-dependent process. BMDCs were derived from WT, NLRP3^−/−^, or MyD88^−/−^ mice and stimulated with media, glucopyranosyl lipid adjuvant-aqueous formulation (GLA-AF) (4 µg/mL), or SE for 5 h. Cells were lysed and assayed for pro-IL-1β by ELISA. (**C**) SE stimulates IL-1β production in a TLR4-dependent process. BMDCs were derived from WT, TLR2^−/−^, or TLR4^−/−^ mice and stimulated with media, LPS, or SE for 5 h. Cells were lysed and assayed for pro-IL-1β by ELISA. (**D**) TLR4^−/−^ mice are not generally immune deficient. BMDCs were derived from WT or TLR4^−/−^ mice and stimulated with media, FSL-1 (100 µg/mL), or SE. Cells were lysed and assayed for pro-IL-1β by ELISA. Data are representative of 3–6 experiments with similar results, showing mean ± SEM; statistical significance was determined via one-way ANOVA followed by a Tukey test for multiple comparisons. *** *p* < 0.001.

**Figure 2 pharmaceutics-14-02557-f002:**
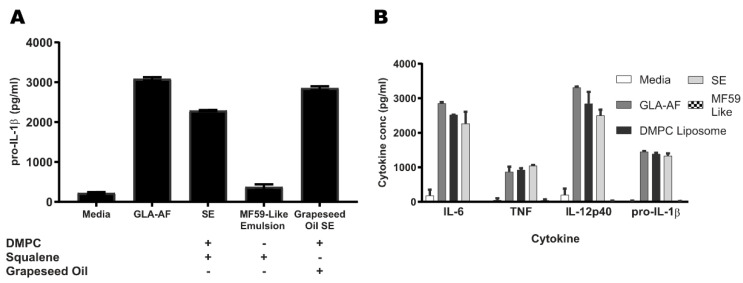
1,2-dimyristoyl-sn-glycero-3-phosphocholine (DMPC) is necessary and sufficient for pro-IL-1β and cytokine production. WT BMDCs were stimulated with media, GLA-AF, squalene SE, MF59-like emulsion, or grapeseed oil SE for 5 h. (**A**) Pro-IL-1β production is ablated in DMPC-free emulsions. Lysate was collected and assayed for pro-IL-1β by ELISA, using the formulations marked below the axis. (**B**) SE and GLA have similar cytokine profiles. WT BMDCs were stimulated as in (**A**) or with DMPC prepared as a liposome for 5 h. Lysates were assayed for pro-IL-1β, IL-6, TNFα, and IL-12p40 by ELISA. Data are shown as the mean +/− SEM of *n* = 3–6 replicates. Data are representative of three experiments with similar results.

**Figure 3 pharmaceutics-14-02557-f003:**
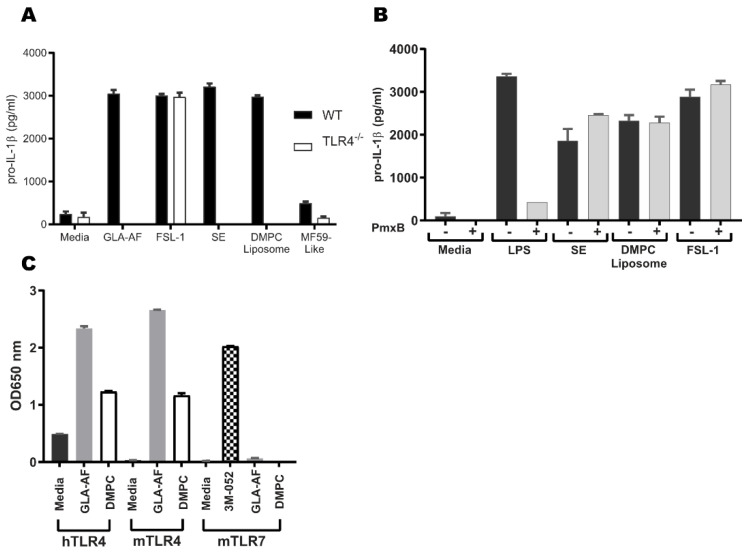
DMPC activates human and murine TLR4 pathway. (**A**) DMPC recapitulates the pro-IL-1β production of SE in a TLR4-dependent manner. WT and TLR4^−/−^ BMDCs were stimulated with media, GLA, SE, DMPC liposomes, or MF59-like emulsion for 5 h. Lysate was collected and assayed for pro-IL-1β by ELISA. (**B**) SE and DMPC stimulation of pro-IL-1β production is not a result of endotoxin contamination. WT BMDCs were stimulated with media, LPS, SE, DMPC liposomes, or MF59-like emulsion pretreated with polymyxin B (PmxB) for 5 h. Lysate was collected and assayed for pro-IL-1β by ELISA. (**C**) DMPC has activity in both human and murine TLR4s. HEK reporter cells transfected with murine (m) or human (h) TLR4 or murine TLR7 were stimulated with media, GLA-AF, DMPC liposomes, or 3M052-AF (TLR7/8 ligand) for 44 h. The production of SEAP induced by NF-κB and AP-1 activation triggered by TLR4 or TLR7 ligand stimulation was measured at OD 650. Data are shown as the mean +/− SEM of a representative with each condition performed in triplicate. Data are representative of three experiments with similar results.

**Figure 4 pharmaceutics-14-02557-f004:**
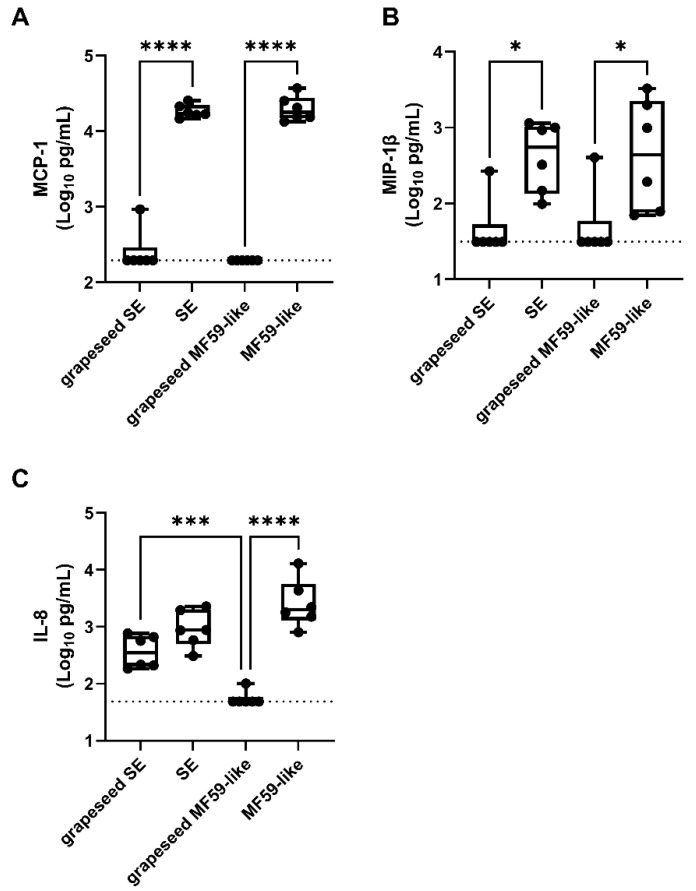
Human whole blood response to emulsion formulations is dominated by oil-phase component. In vitro stimulation of human whole blood with oil-in-water emulsions at 0.4% *v*/*v* oil. Squalene-containing formulations led to increased secretion of (**A**) MCP-1, (**B**) MIP-1β, and (**C**) IL-8. Log-transformed cytokine concentrations were measured in supernatants of heparinized blood stimulated by incubation with emulsions. Box and whisker plots indicate 1st–3rd quartiles with whiskers representing the minimum and maximum values from *n* = 6 donors (3 male and 3 female). Statistical significance was determined by one-way ANOVA followed by Tukey’s correction for multiple comparisons. * *p* < 0.05, *** *p* < 0.001, and **** *p* < 0.0001.

**Figure 5 pharmaceutics-14-02557-f005:**
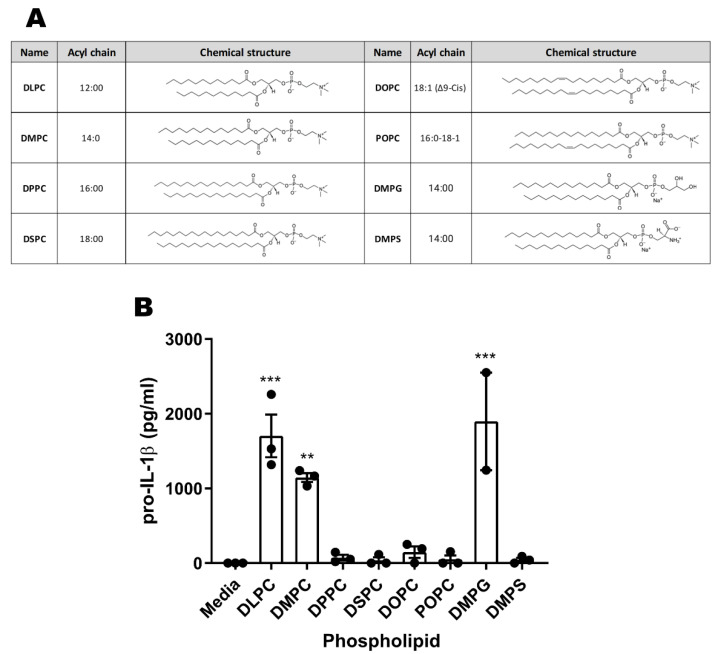
Both the acyl chain length and headgroup structure affect the activity of murine BMDCs. WT BMDCs were stimulated with a panel of phospholipids (**A**) as liposomes. (**B**) Stimulation of pro-IL-1β production is dependent on lipid structure. After 5 h, lysates were collected and assayed for pro-IL-1β by ELISA. Data are shown as the mean +/− SEM of *n* = 3 replicates. Data are representative of three experiments with similar results, showing mean ± SEM. Statistical significance was determined via one-way ANOVA followed by a Dunnett correction for multiple comparisons to the media-only group. ** *p* < 0.01 and *** *p* < 0.001.

**Figure 6 pharmaceutics-14-02557-f006:**
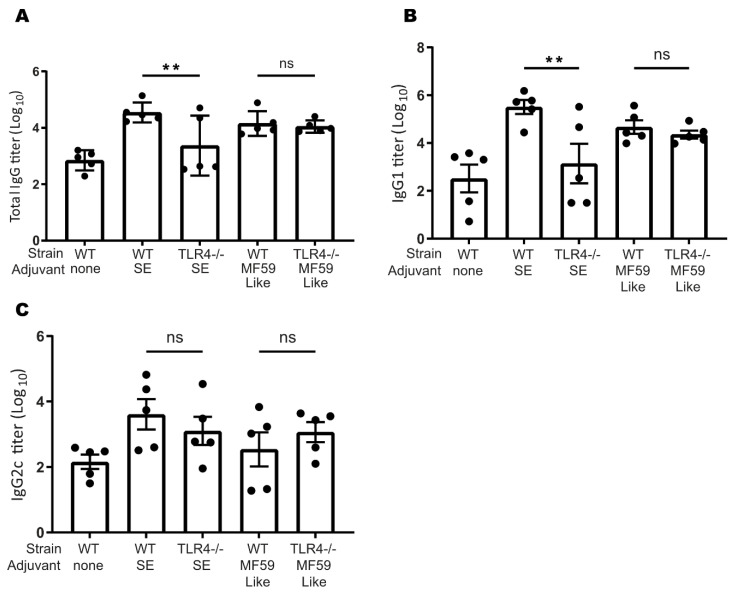
SE acts through TLR4 to augment humoral responses to vaccination. DMPC-containing adjuvants act in a TLR4-dependent process, whereas MF59-like emulsions show TLR4-independent activity. WT and TLR4^−/−^ mice were immunized once with a tuberculosis protein antigen (ID97) either unadjuvanted or adjuvanted with SE or MF59-like emulsion. Three weeks later, peripheral blood was collected. Serum was assayed for antigen-specific (**A**) IgG, (**B**) IgG1, and (**C**) IgG2c by ELISA. Data are shown as the mean +/− SEM of a representative population with each condition performed in triplicate. Data are pooled from two experiments with similar results. ** *p* < 0.01 and ns = not significant.

## Data Availability

The data presented in this study are available on request from the corresponding author.

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
