# Peer review of "Formulated Phospholipids as Non-Canonical TLR4 Agonists"

_pharmaceutics, 2022, doi:10.3390/pharmaceutics14122557_

Round 1
Reviewer 1 Report
This is an important study, I recommend it for publication after the following points are addressed.
1. The mechanism is still not clear that DMPC has such a specific function.
2. What are the sizes and zeta potential of the emulsions and liposomes used in this study?
3. The resolution of figure 4 and 5 should be improved to a higher level.
4. Line 418-421, several studies (ACS Nano 2021, 15, 9, 14022–14048; Polymers 2021, 13, 3395) should be included to support such a claim.
Reviewer 2 Report
Dear Author,
Your scientific work "Formulated phospholipids as non-canonical TLR4 agonists" is inteeting and important. Accept in present form!
Reviewer 3 Report
The authors confirm that the phospholipid DMPC, a formulation component of oil-in-water vaccine adjuvant emulsion, function as a no-canonical TLR4 agonist in in vitro models of murine and human TLR4 and murine models of TLR4 activity. The research design is appropriate. The data are sounded and support the conclusions. These results will be helpful to the elucidation of adjuvant-emulsion action mechanisms and the interactions between formulation components and the innate immune responses.
Minor comment:
1. As mentioned in the Introduction section, the combination adjuvants with TLR agonists can promote CD4+ T cell responses. But in the recent development of T cell epitope peptides vaccines of SARS-CoV-2, the TLR agonists also were used as adjuvants to promote CD8+ T cell responses and CD4+ T cell responses, but not antibody production. This mention will make the background more comprehensive.
2. For the intracellular accumulation of pro-IL-1β, more visual detection method is intracellular cytokine staining and flow cytometry. Why only ELISA was used in this study?
